# Exploring Enhanced Hydrolytic Dehydrogenation of Ammonia Borane with Porous Graphene-Supported Platinum Catalysts

**DOI:** 10.3390/molecules29081761

**Published:** 2024-04-12

**Authors:** Zhenbo Xu, Xiaolei Sun, Yao Chen

**Affiliations:** 1The State Key Laboratory of Refractories and Metallurgy, Faculty of Materials, Wuhan University of Science and Technology, Wuhan 430081, China; 2School of Materials Science and Engineering, Nankai University, Tianjin 300350, China

**Keywords:** graphene, nanodot, porous, platinum, catalyst, ammonia borane

## Abstract

Graphene is a good support for immobilizing catalysts, due to its large theoretical specific surface area and high electric conductivity. Solid chemical converted graphene, in a form with multiple layers, decreases the practical specific surface area. Building pores in graphene can increase specific surface area and provide anchor sites for catalysts. In this study, we have prepared porous graphene (PG) via the process of equilibrium precipitation followed by carbothermal reduction of ZnO. During the equilibrium precipitation process, hydrolyzed N,N-dimethylformamide sluggishly generates hydroxyl groups which transform Zn^2+^ into amorphous ZnO nanodots anchored on reduced graphene oxide. After carbothermal reduction of zinc oxide, micropores are formed in PG. When the Zn^2+^ feeding amount is 0.12 mmol, the average size of the Pt nanoparticles on PG in the catalyst is 7.25 nm. The resulting Pt/PG exhibited the highest turnover frequency of 511.6 min^−1^ for ammonia borane hydrolysis, which is 2.43 times that for Pt on graphene without the addition of Zn^2+^. Therefore, PG treated via equilibrium precipitation and subsequent carbothermal reduction can serve as an effective support for the catalytic hydrolysis of ammonia borane.

## 1. Introduction

Hydrogen gas serves as a clean energy source, boasting a specific density of 143 MJ kg^−1^, significantly surpassing that of conventional fossil fuels [1,2]. However, the transport and storage of hydrogen are critical barriers hindering the development of the hydrogen economy. Chemical hydrogen storage materials, such as ammonia borane (AB) [3,4], formic acid [5,6] and sodium borohydride [7], offer mild hydrogen release and relatively safe transport, presenting significant potential for large-scale practical applications of hydrogen energy [8,9]. Among chemical hydrogen storage materials, non-toxic and stable AB can exhibit an exceptionally high hydrogen storage amount and density (19.6 wt% and 146 g L^−1^) [10]. It is one of the most competitive candidates among chemical hydrogen storage materials [11]. The release of hydrogen from AB typically involves thermolysis, alcoholysis and hydrolysis processes [12,13,14]. Thermolysis and alcoholysis often require high-temperature operation over catalysts and may generate organic byproducts. In contrast, hydrolysis is the simplest method, only requiring the assistance of a catalyst under ambient conditions [15].

Efficient catalysts for the hydrolysis of AB involve transition metals, such as Pt, Rh, Pd and Ru, among which the precious metal Pt catalyst demonstrates the optimal catalytic activity [16,17]. However, the low abundance and high cost of Pt hinder its widespread application [18,19,20]. In particular, when existing independently, Pt particles tend to aggregate, which reduces the amount of exposed atomic surfaces, ultimately leading to low catalytic activity and high costs. Therefore, supports have been exploited to immobilize homogenously dispersed Pt catalysts, thereby promoting Pt atomic utilization. Nanoscale porous materials, such as porous carbon, mesoporous silica and metal–organic frameworks (MOFs), serve as ideal carriers for encapsulating ultrafine catalysts to prevent sintering [21].

Graphene, a two-dimensional (2D) monolayer of graphite with a theoretical specific surface area (SSA) of 2630 m^2^ g^−1^ [22,23,24], is also used as a support for catalysis [25]. Although restacking of solid graphene largely decreases the practical SSA, activated graphene derived from microwave-exfoliated graphite oxide (GO) resulted in a large SSA of 3100 m^2^ g^−1^ [26]. However, the pore distribution of the activated graphene is difficult to control during KOH activation. As GO has many oxygen-functional groups, the pore structures in graphene can be constructed at the microscopic level via reactions with oxygen or adjacent carbon atoms. Based on this, H_2_O_2_ was used to etch the carbon atoms around the oxygen defect sites of GO via hydrothermal treatment. The resulting porous graphene aerogel possessed an SSA of 830 m^2^ g^−1^, while the SSA of graphene without the H_2_O_2_ treatment was only 260 m^2^ g^−1^ [27]. However, a wide pore distribution still cannot be avoided. The carbothermal reaction presents an alternative method for generating in-plane pores in graphene. By subjecting a mixture of graphene oxide (GO) and oxometalates in water to freeze drying, followed by annealing in a nitrogen environment, carbon atoms from GO reacted with metal oxide nanoparticles formed in situ from oxometalates. Subsequent treatment with acid removed the metal-containing species, leaving behind nano-scaled pores ranging from 1 to 50 nm on the graphene sheets [28]. However, the random distribution of oxometalates on GO caused the wide pore distribution in graphene. Catalytic gasification of adjacent carbon atoms in GO with SnO_2_ produced graphene nano meshes with an SSA of 261 m^2^ g^−1^ [29]. The size distribution could be controlled only in the range between 10 and 20 nm. Similarly, Ag nanoparticles were in situ generated from silver acetate, which was mixed with GO, catalyzing the gasification of carbon atoms in GO with the aid of copper wire under microwave irradiation. Upon adding the appropriate amount of silver acetate into the GO dispersion, nanopores with an average size of 5 nm were produced [30]. The limitation of this catalytic gasification approach for generating pores in graphene resides in the costly nature of Ag. Microwave treatment was similarly employed for pore formation in slightly oxidized graphite intercalation compounds. Iterative microwave treatments could realize the fabrication of holey graphene nanoplatelets [31]. It was also observed that annealing commercial graphene in air could induce the formation of micropores with a diameter of 1.9 nm and mesopores measuring 12 nm. Prolonged oxidation in air led to an increase in the proportion of mesopores. During the initial transition of annealing in air, gasification of defective carbons resulted in the creation of pores, although the resulting material remained structurally and chemically similar to the original commercial graphene. The fraction of mesopores and the oxygen content continued to rise with increasing temperature until a second transition occurred, where the gasification of graphitic carbons occurred due to over oxidation [32]. Annealing in air is a straightforward method; however, achieving a narrow pore distribution primarily concentrated in the micropore range is still challenging.

In this study, we anchored the ZnO nanodots on GO in a mixed solution of water and N,N-dimethylformamide (DMF) by equilibrium precipitation in an oil bath, and then utilized the carbothermal reduction of ZnO with carbon in graphene to produce nanoscale porous graphene (PG), yielding a narrow pore distribution centered in the micropore range. This approach appears to address the limitations of the aforementioned methods for pore creation in graphene. When Zn^2+^ was 0.12 mmol, the average size of the ZnO nanodots was estimated to be less than 1 nm. The resulting PG-0.12 showed a narrow pore distribution around 1 nm with an SSA of 102 m^2^ g^−1^, which was beneficial for immobilizing the highly dispersed nanoscale catalysts. Mild aggregations of the Pt nanoparticles were deposited around the pores in the basal plane of graphene. The average length of the mild Pt aggregations on PG-0.12 was 7.25 nm. The Pt catalysts loading on PG-0.12 exhibited the optimal catalytic performance of hydrogen release from AB with a turnover frequency (TOF) value of 511.6 mol_H_2__ mol_Pt_^−1^ min^−1^, which is 2.5 times that of the Pt catalysts on the graphene without adding Zn^2+^.

## 2. Results and Discussion

As shown in Figure 1, the Pt/PG catalyst was prepared by loading platinum onto PG via three steps: anchoring ZnO nanodots on GO by precipitation of Zn^2+^, heat treatment to produce pores in the basal plane of PG and immobilization of Pt nanoparticles on PG. During the first step of anchoring ZnO nanodots on GO, DMF was sluggishly hydrolyzed at 95 °C to produce hydroxyl groups that could react with Zn^2+^ in an equilibrium state to produce ZnO nanodots. Simultaneously, GO was reduced to give rise to RGO with the in situ generated hydroxyl groups at the elevated temperature in the oil bath. The ZnO nanodots were prone to being anchored on the defects of RGO [33]. Heat treatment at 900 °C in the inert gas guaranteed that a reduction reaction between the ZnO nanodots and carbon atoms in RGO occurred, during which the resulting metallic Zn was evaporated [34]. After the heat treatment, the powder was further subjected to washing with acid to completely remove residual Zn or ZnO. Consequently, nanopores were formed in PG. Finally, Pt^2+^ thoroughly mixed with the PG aqueous dispersion was reduced with NaBH_4_ to obtain the Pt/PG catalysts.

Figure 2a shows the XRD patterns of GO, ZnO/RGO-0.12 and PG-0.12. The peak for GO at 11.5° demonstrates that the distance of the (002) plane was 0.769 nm, larger than 0.336 nm for the (002) plane of pristine graphite, indicating that oxygen-functional groups were produced on the graphitic basal planes. In the XRD pattern of ZnO/RGO-0.12, the peak for GO at 11.5° disappeared, instead, the corresponding broad (002) peak was centered at 25.36°, indicating the reduction of GO. The intensive peak well corresponded to SiO_2_ (JCPDS: 89-1961) which was from the impurity of the glassware. After heat treatment and acid washing, the XRD pattern of PG showed an obvious peak at 26.6° attributed to the (002) plane of graphite. It is concluded that GO was reduced to RGO at the elevated temperature in the oil bath, which was further reduced to restacking graphene after heat treatment at 900 °C.

It is worth noting that no ZnO peaks could be seen at all in the XRD pattern of ZnO/RGO-0.12, which may be arising from the very small amount of Zn^2+^ in the feed. The ICP-MS analysis determined that the Zn weight content in ZnO/RGO-0.12 was 6.5%. To further demonstrate the transformation of Zn^2+^ in the oil bath, the feeding amount of Zn^2+^ increased to 1.2 and 2.4 mmol. The resulting materials were examined by XRD again (Figure 2b). When the Zn^2+^ feeding amount was 1.2 mmol, no evident peaks were found for ZnO/RGO-1.2. Annealing of ZnO/RGO-X at 300 °C for 1 h in Ar was used to produce a crystalline sample of ZnO/RGO-X-300. ZnO/RGO-1.2-300 exhibited distinct characteristic peaks at 31.73°, 34.5° and 36.2°, corresponding to the (100), (002) and (101) crystal planes of ZnO (JCPDS: 75-1526), respectively [35]. Heat treatment increased the peak intensities of ZnO and decreased the full width at half maximum, indicating that crystallinity of ZnO improved after heat treatment. When the feeding amount of Zn^2+^ increased to 2.4 mmol, the inconspicuous peaks of ZnO could be seen in the XRD pattern of ZnO/RGO-2.4, suggesting that the amorphous ZnO was formed during the oil bath. Similarly, after annealing at 300 °C, ZnO/RGO-2.4-300 also possessed the clearer characteristic peaks of ZnO. According to the Scherrer formula, the grain sizes of ZnO/RGO-1.2-300 and ZnO/RGO-2.4 were calculated to be 5.3 and 4.7 nm, respectively. This suggests that ZnO nanodots were generated in the mixture of DMF and water at elevated temperatures during the oil bath process, provided that the feeding amount of Zn^2+^ was less than 2.4 mmol per 100 mg of the GO precursors. It is also hypothesized that the grain size of the ZnO nanodots in ZnO/RGO-0.12 must be smaller than 4.7 nm. Therefore, it is concluded that zinc acetate underwent the hydrolysis reaction in equilibrium with the hydroxyl groups from the hydrolyzed DMF during refluxing in the oil bath to produce amorphous ZnO nanodots.

Figure 2c shows the XPS survey spectra of GO, ZnO/RGO-0.12 and PG-0.12. The XPS spectrum of ZnO/RGO-0.12 exhibited the C, O and Zn atomic contents were 73.76%, 25.4% and 0.83%, determining that the Zn weight content was 4%, which is close to the ICP-MS result (6.5%). The C/O ratios of GO, ZnO/RGO-0.12 and PG-0.12 were 2.26, 2.9 and 22.51, respectively. The significant increase in the C/O ratio from GO to PG-0.12 proves that the most oxygen-functional groups in GO were eliminated during the heat treatment at 900 °C. According to the widely recognized Lerf–Klinowski model [36], GO contains functional groups, involving hydroxyl and cyclic ether groups on the basal plane and carboxyl groups and carbonyl groups at the edge. Figure 2d–f shows the C 1s spectra of GO, ZnO/RGO-0.12 and PG-0.12, respectively. In these C 1s spectra, the fitting peaks at the binding energies of 284.6, 286.6, 287.8 and 288.8 eV corresponded to C–C, C–O, C=O and COOH, respectively [37,38,39]. The intensities of the C–O peaks impressively decreased from GO to ZnO/RGO-0.12 and PG-0.12. Similar changes occurred for the C=O and COOH groups. Especially, the COOH groups almost completely disappeared. It can be indicated that GO was reduced to RGO with significant decrease in these oxygen-functional groups after the solvothermal treatment. The subsequent high-temperature heat treatment further removed these oxygen-containing functional groups. Figure 2g shows the XPS Zn 2p spectrum of ZnO/RGO-0.12, where the fitting peaks at 1022.3 and 1045.4 eV corresponded to Zn 2p_3/2_ and Zn 2p_1/2_, respectively. The binding energy difference of 23 eV between Zn 2p_3/2_ and Zn 2p_1/2_ reveals that the oxidation state of ZnO/RGO-0.12 was Zn^2+^, inferring that ZnO was produced after the solvothermal treatment in the oil bath.

The Raman spectra of GO, ZnO/RGO-0.12 and PG-0.12 are shown in Figure 2h. The D peaks at about 1344 cm^−1^ reflected local lattice defects and disorder of the carbon materials, while the G peaks at about 1585 cm^−1^ corresponded to ordered sp^2^-hybridized carbon atoms of the graphene sheets. The peak intensity ratio of the D peak to the G peak (*I*_D_/*I*_G_) is commonly used to evaluate the defect density of graphene. The *I*_D_/*I*_G_ ratios of GO, ZnO/RGO-0.12 and PG-0.12 were 1.01, 1.06 and 1.27, respectively. The increase in the *I*_D_/*I*_G_ ratio from GO to ZnO/RGO-0.12 resulted from an increase in the defect density, which is attributed to the produced defects after the removal of some oxygen-functional groups and the covalent binding of the amorphous ZnO nanodots and oxygen-functional groups on the basal plane of graphene. The significant increase in defect density in PG may be attributed to the formation of pores by the carbothermal reaction between the ZnO nanodots and adjacent carbon atoms in graphene and the defects left after the further removal of oxygen-functional groups.

The N_2_ adsorption and desorption isotherms of PG-0.06, PG-0.12 and PG-0.24 are shown in Figure 2i. The SSAs of PG-0.06, PG-0.12 and PG-0.24 were 136, 102 and 92 m^2^ g^−1^, respectively. As the Zn^2+^ feeding amount increased, the absorbed quantities in the high relative pressure regions decreased, suggesting that the number of the mesopores decreased, which is responsible for a gradual decrease in SSA of PG-Xs. As expected, the evident volume expansion of the materials after heat treatment could be observed in PG-0 without adding Zn^2+^. However, the volume expansion was gradually inhibited with the increased Zn^2+^ feeding amount. On the one hand, the ZnO nanodots anchoring on the graphene increased the weight of the materials. On the other hand, the oxygen-functional groups were covalently bonded to the ZnO nanodots. The two factors inhibited the volume expansion of the graphene which was supposed to be from the unanchored oxygen escaping from GO during the heat treatment process. Consequently, the suppression of the volume expansion decreased the exposed mesopores from the channel between graphene layers. According to the pore distribution shown in the inset of Figure 2i, the maximum d*V*/d*D* of PG-0.06, PG-0.12 and PG-0.24 lied at 1.09, 1.18 and 1.27 nm, respectively. The maximum d*V*/d*D* values at the micropores in the three samples were the orders of magnitude larger than those at the mesopores, suggesting that the carbothermal reaction produced narrow pore distributions in PG-X. PG-0.12 showed the narrowest pore distribution. The formation of micropores centered at 1.18 nm and absence of small mesopores less than 4 nm are beneficial for immobilizing highly dispersed Pt nanoparticles.

Figure 3a shows the TEM image of ZnO/RGO-0.12, where no evident metal particles could be seen on 2D RGO sheets. Combining the XRD, XPS and ICP-MS analyses which demonstrated the presence of Zn element in the ZnO/RGO-0.12 sample, the ZnO absence in Figure 3a may be because the amorphous small ZnO particles, due to the ultralow Zn^2+^ feeding amount, were ejected by the high-energy electron beam during TEM operation. By increasing the Zn^2+^ feeding amount to 0.6 mmol, uniformly dispersed ZnO particles could be found on RGO, as shown in Figure 3b,c. The average grain size of ZnO was 1.17 nm, as shown in the inset of Figure 3b. With a greater Zn^2+^ feeding amount of 1.2 mmol, the average grain size of the ZnO nanodots with more obvious profiles, increased to 1.57 nm (Figure 3d,e). According to the high-resolution TEM image of ZnO/RGO-1.2 in Figure 3f, the lattice fringe of the single nanodot was 0.26 nm, corresponding to the (002) plane of ZnO. Figure 3g shows the C, O and Zn elemental mappings of ZnO/RGO-0.6, demonstrating the uniform distribution of the ZnO nanodots in the sample. Compared with those in ZnO/RGO-1.2, which accounted for 25.7%, the nanodots less than 1.2 nm in diameter in ZnO/RGO-0.6 constituted 58.8%, suggesting that reducing the Zn^2+^ feeding amount resulted in a decrease in the particle size of ZnO. Therefore, it can be deduced that the size of the ZnO nanodots in ZnO/RGO-0.12 would be smaller than 1 nm. After heat treatment, uniformly dispersed pores less than 2 nm could be clearly seen on the surface of PG-0.12, as shown in Figure 3h,i, which were formed by the carbothermal reduction reaction of ZnO.

In Figure 4a, the XRD pattern of Pt/PG-0.12 revealed a faint Pt (111) crystal plane peak at 39.75°, suggesting that Pt^2+^ ions were reduced to the metallic Pt particles with NaBH_4_. Figure 4b shows the XPS survey spectrum of Pt/PG-0.12, which displays characteristic peaks of C, O and Pt. The atomic percentages of C, O and Pt elements were 94.86%, 4.78% and 0.36%, respectively, corresponding to a Pt mass loading of 5.47%. As shown in Figure 4c, the fitting peaks at 71.55 and 74.9 eV in the XPS Pt 4f spectrum corresponded to the Pt^0^ 4f_7/2_ and 4f_5/2_ peaks, while the fitting peaks at 72.9 and 76.25 eV corresponded to the Pt^2+^ 4f_7/2_ and 4f_5/2_ peaks. The fitting peaks corresponding to Pt^0^ dominated in the Pt 4f spectrum of Pt/PG-0.12. The presence of Pt^2+^ infers that the partial Pt atoms were immobilized to C atoms by the Pt–C bond in PG-0.12 [40]. The predominance of metallic Pt^0^ in the XPS spectrum, combining with the XRD analysis, demonstrates the reaction of the Pt^2+^ precursors to the metallic Pt was realized with NaBH_4_. Figure 4d,e depict the TEM images of the Pt/PG-0.12 catalyst. Mild aggregations of the Pt nanoparticles could be found around the pores in the basal plane of graphene. The average length of the mild Pt aggregations on PG-0.12 was 7.25 nm. The lattice fringes with an interplanar spacing of 0.225 nm in a monodispersed Pt nanoparticle corresponded well to the Pt(111) crystal plane (JCPDS: 01-1194). Figure 4f shows the TEM image of the control Pt/PG-0 catalyst. More severe aggregations of the Pt nanoparticles were grown on graphene. The average length of the severe Pt aggregations was 11.53 nm. It is worth noting that the range of the Pt aggregation in Pt/PG-0 was from 2.21 to 40.52 nm. The intensive contrast indicates that the pores in Pt/PG-0.12 help overcome the severe aggregations of the Pt nanoparticles.

Figure 4g illustrates the hydrogen evolution rates of the AB hydrolysis over the Pt/PG-X catalysts with different Zn^2+^ feeding amounts. A total of 152 mL of H_2_ was released within 1.25 min over Pt/PG-0.12, which exhibited the highest catalytic performance among all the catalysts. Conversely, the control Pt/PG-0 catalyst completely released 154 mL of H_2_ from AB within 3 min. Figure 4h shows the TOF values of the Pt/PG-X catalysts. The control Pt/PG-0 catalyst exhibited a TOF value of only 210.9 min^−1^. Pt/PG-0.12 delivered the highest TOF value of 511.6 min^−1^, which is 2.53-fold that of the control Pt/PG-0 catalyst. The Pt catalysts on commercial XC-72R and YP-50f possessed TOF values of 448.2 and 417.7 min^−1^, respectively, which was indicated as the dash lines in Figure 4h. Although YP-50f has a much larger SSA (1944 m^2^ g^−1^) [41], the TOF value of Pt/YP-50f was only comparable to Pt/XC-72R. The superior performance of Pt/PG-0.12 compared with that of Pt on commercial carbon supports indicates that the Pt/PG catalysts are promising for various applications in catalysis. Overall, the TOF value initially increased and then decreased with increasing Zn^2+^ feeding. When the quantity of Zn^2+^ was minimal, such as 0.06 mmol, ZnO/RGO-0.06 showed a broad range of pore sizes spanning from micropores to mesopores, which was supposed to fail to effectively prevent the aggregation of Pt nanoparticles. Conversely, with a higher Zn^2+^ feeding amount, for instance 0.24 mmol, ZnO/RGO-0.24 exhibited a broader pore size distribution primarily within the micropores range and a reduced SSA. The decline in SSA did not facilitate the even dispersion of Pt^2+^ across the surface of individual graphene layers. When the Zn^2+^ feeding amount was 0.12 mmol, the narrowest pore distribution with a relatively moderate specific surface area was achieved, allowing Pt-PG-0.12 to exhibit the optimal catalytic performance of the AB hydrolysis. These results indicate that the pores precisely controlled by carbothermal reduction prevented the severe agglomeration of Pt particles and hence significantly promoted the catalytic activity. Figure 4i shows the cyclic stability of H_2_ released from AB over Pt/PG-0.12. The performance of the catalyst gradually worsened, mainly due to the ongoing crystallization and agglomeration of Pt particles after cycling, a phenomenon similar to what had been reported for Pt on carbon nanotubes [42] or graphene [43].

## 3. Experimental

### 3.1. Materials Preparation

#### 3.1.1. Preparation of GO

GO was prepared by the modified Hummer’s method [44]. 2 g of natural graphite (325 mesh), 2 g of NaNO_3_ and 92 mL of H_2_SO_4_ were mixed and held at −1 to −2 °C for 19 h in a thermostatic tank. After that, 12 g of KMnO_4_ was slowly added and stirred at room temperature for 4 h. After slowly adding 184 mL of water, the suspension was heated at 98 °C for 30 min in a water bath. Subsequently, the suspension was treated with 400 mL of water (60 °C) and 40 mL of H_2_O_2_ and stirred for 10 min. Then the mixture was centrifuged and transferred to a beaker, in which 40 mL of HCl was added and stirred for 10 min. The product was further washed with water by centrifugation until the volume of the GO colloids no longer expanded. Finally, the GO colloids were transferred to a petri dish and dried at 60 °C for 48 h to obtain dried GO sheets.

#### 3.1.2. Preparation of PG

A total of 100 mg of GO was dispersed in 50 mL of deionized water by ultrasonication until homogeneous colloids formed. Subsequently, X mmol of Zn(CH_3_COO)_2_·2H_2_O was dissolved in 50 mL of water and added to the GO dispersion in the flask which was placed in an oil bath at 95 °C. After stirring for 5 min, 100 mL of DMF was added. The mixture was kept in an oil bath for 5 h. Then, the sediment was sequentially washed with alcohol and water, followed by drying, giving rise to amorphous ZnO on RGO (ZnO/RGO-X). ZnO-RGO-Xs were heated to 900 °C in an argon atmosphere at a rate of 5 °C min^−1^ and maintained for 1 h, followed by acid washing and water rinsing to obtain PG-Xs. The control sample with neither Zn^2+^ nor DMF was named PG-0.

#### 3.1.3. Preparation of Pt/PG

Then, 0.1 g of PG-X was sonicated in 2.5 mL of water for 10 min, 31 μL of 0.32 mol of K_2_PtCl_4_ was added to the PG dispersion and stirred for 1 h. After 1 h, 0.05 g of NaBH_4_ dissolved in 1 mL of water was rapidly added and stirred for another 1 h. Finally, the precipitate was washed to neutrality, producing Pt/PG-X. For Pt/PG-0, all the steps were the same, but 15 mL of water was necessary to submerse PG-0.

### 3.2. Characterizations

X-ray diffraction (XRD) was performed on an X-ray diffractometer (SmartLab SE, Rigaku, Tokyo, Japan) using Cu Kα radiation (λ = 1.5406 Å). Detailed microstructural and compositional analyses were conducted using transmission electron microscopy (TEM, JEM-F200, JEOL, Tokyo, Japan) with energy-dispersive X-ray spectroscopy (EDS) analysis. The suspensions by ultrasonically dispersing the samples for TEM in ethanol were dripped onto Cu meshes and dried. X-ray photoelectron spectroscopy (XPS) was carried out using an Al Kα source on K-Alpha (Thermo Fisher Scientific, Waltham, MA, USA) to detect surface elements and chemical states. The powder samples were pressed into the tablet prior to the XPS detections. Raman spectroscopy was employed to characterize the disorder in the materials utilizing HR Evolution (HORIBA, Tokyo, Japan) with a laser with a wavelength of 532 nm. Inductively coupled plasma mass spectrometry (ICP-MS) was utilized to determine the metal loading of the materials using Agilent 7800 (Agilent, Santa Clara, CA, USA). The ICP-MS pretreatment adopted the digestion/dissolution method, and the calibration was carried out according to “Quadrupole inductively coupled plasma mass spectrometer calibration specification (jjf 1159-2006)”. The SSA and pore size distribution of the materials were analyzed using an ASAP 2460 3.01 (Micromeritics, Norcross, GA, USA).

### 3.3. Catalytic Performance

The hydrolysis reaction of AB to produce H_2_ is shown in Equation (1). The performance of the catalysts was tested by the drainage method (Figure 5). Pt/PG-X was mixed with water to form a dispersion which was transferred to a two-necked flask placed in a water bath at 30 °C. The total volume of the dispersion depended on the transfer capability of the different catalysts. Specifically, 2 mL dispersions were employed for Pt/PG-0.6 and Pt/PG-1.2; 4 mL for Pt/PG-0.24, Pt/PG-0.12, Pt/PG-0.06, Pt/PG-0.012; 10 mL for Pt/PG-0. After sealing the flask, 2 mol AB in 2 mL water was injected into the flask using a syringe. The stopwatch timing and the marked gas scale were used to record simultaneously until no bubbles are generated. For the durability test, the catalysts were recycled by washing with water after each cycle. The reaction rates of the catalysts were evaluated by TOFs, which can be calculated using Equation (2):(1)$${NH}_{3}{BH}_{3}+{2H}_{2}O={NH}_{4}{BO}_{2}+{3H}_{2}↑$$
(2)$$TOF=\frac{p_{0}V}{{RTn}_{M}t}$$
where *p*_0_ is atmospheric pressure, *V* is the total volume of gas produced in hydrolysis of AB, *R* is molar gas constant, *T* is the reaction temperature, *n*_M_ is the amount of the metal catalyst, *t* is terminal time for the reaction.

## 4. Conclusions

We have prepared the PG materials as catalytic supports by equilibrium precipitation and subsequent carbothermal reduction. During the equilibrium precipitation process, the amorphous ZnO nanodots anchored on the oxygen-functional groups on the graphene basal plane by the sluggish hydrolysis reaction of Zn^2+^ and in situ generated hydroxyl groups from hydrolyzed DMF in the oil bath. Meanwhile, GO was reduced to RGO at the elevated temperature. Decreasing the Zn^2+^ feeding amount can decrease the size of the ZnO nanodots on RGO. The average size of the ZnO nanodots was estimated to be less than 1 nm when the Zn^2+^ feeding amount was 0.12 mmol. After carbothermal reduction of ZnO and RGO, pores formed in the resulting PG. When the Zn^2+^ feeding amount was 0.12 mmol, PG-0.12 achieved the narrowest pore distribution, centered primarily in the micropore range, with a moderate surface area, which was beneficial for immobilizing highly dispersed nanoscale catalysts. The Pt nanoparticles were deposited around the pores of PG-0.12. The average particle size of the mild Pt aggregations was 7.25 nm. The resulting Pt/PG-0.12 catalyst exhibited the highest TOF value of 511.6 min^−1^ for the AB hydrolysis among Pt/PG-Xs, which is 2.43 times that for Pt on graphene without the addition of Zn^2+^ and also higher than those for Pt on commercial carbon supports. Therefore, PGs treated by the equilibrium precipitation and subsequent carbothermal reduction can serve as effective supports for the catalytic hydrolysis of AB. Through continuous efforts to optimize the preparation of PG, there is a growing belief that the PG material can serve as versatile support for immobilizing various nanoscale catalysts for a wide range of catalytic reactions.

## Figures and Tables

**Figure 1 molecules-29-01761-f001:**
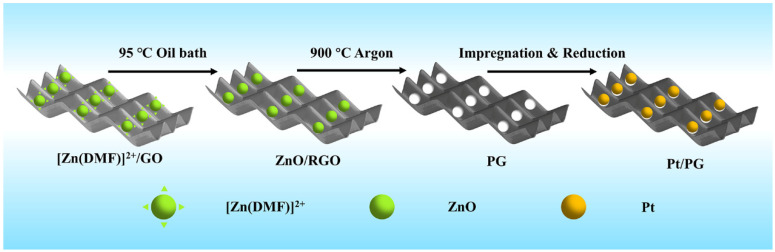
Schematic of Pt/PG preparation.

**Figure 2 molecules-29-01761-f002:**
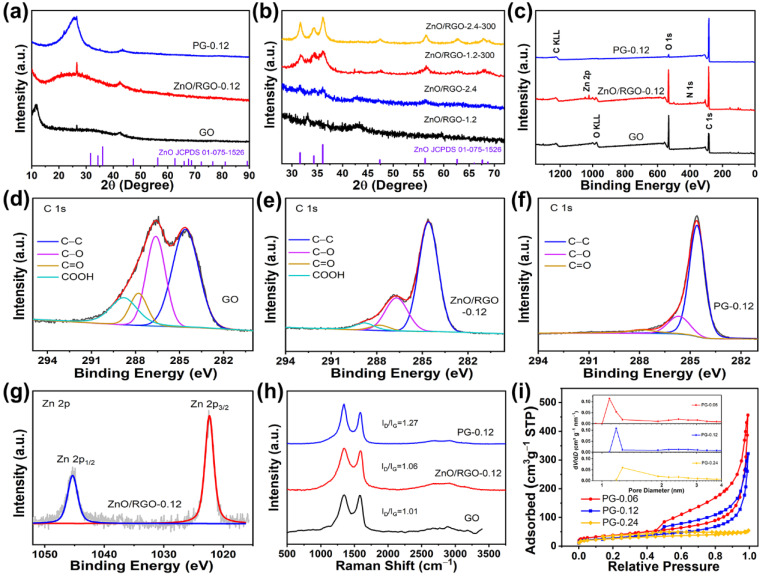
XRD patterns of (**a**) GO, ZnO/RGO-0.12 and PG-0.12 and (**b**) ZnO/RGO-1.2, ZnO/RGO-2.4 and ZnO/RGO-2.4-300, XPS (**c**) survey spectra of GO, ZnO/RGO-0.12 and PG-0.12, C 1s spectra of (**d**) GO, (**e**) ZnO/RGO-0.12 and (**f**) PG-0.12 (red lines for fitting data and black lines for raw data) and (**g**) Zn 2p spectrum of ZnO/RGO-0.12, (**h**) Raman spectra of GO, ZnO/RGO-0.12 and PG-0.12, (**i**) adsorption and desorption isotherms of N_2_ of PG-0.06, PG-0.12 and PG-0.24, inset corresponding to pore distributions.

**Figure 3 molecules-29-01761-f003:**
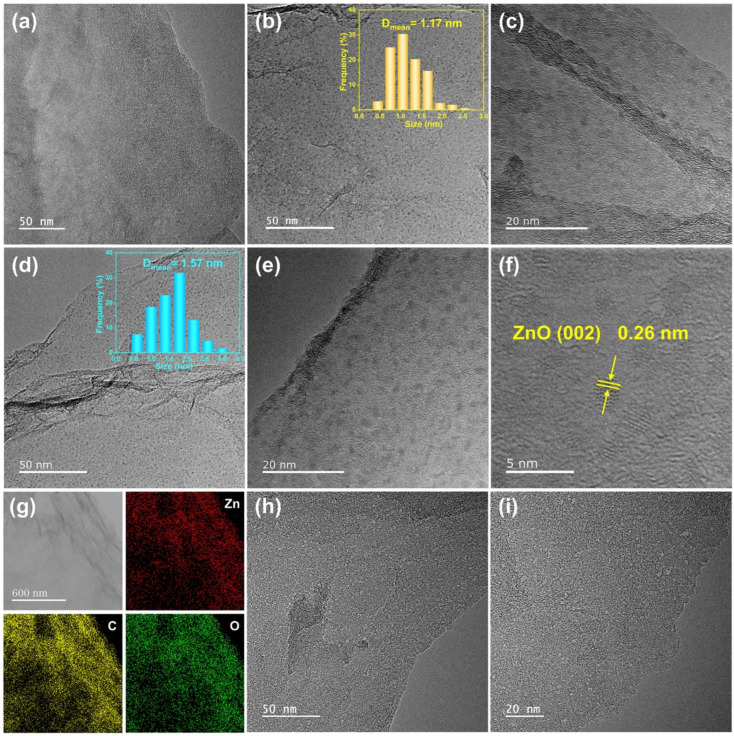
TEM images of (**a**) ZnO/RGO-0.12, (**b**) and (**c**) ZnO/RGO-0.6 and (**d**–**f**) ZnO/RGO-1.2, (**g**) elemental mappings of ZnO/RGO-0.6, (**h**,**i**) TEM images of PG-0.12.

**Figure 4 molecules-29-01761-f004:**
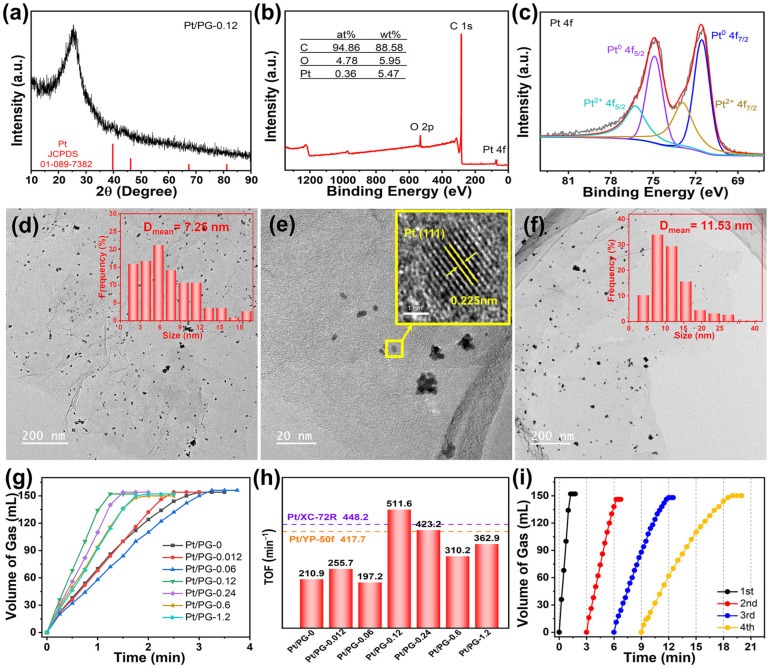
(**a**) XRD pattern, XPS (**b**) survey and (**c**) Pt 4f spectra (red lines for fitting data and black lines for raw data) of Pt/PG-0.12, TEM images of Pt/PG-0.12 at (**d**) low, (**e**) medium and inset of (**e**) high magnification and (**f**) Pt/PG-0 at low magnification, (**g**) hydrogen evolution rates and (**h**) TOFs of AB hydrolysis over Pt/PG-X, Pt/YP-50f and Pt/XC-72R, (**i**) cyclic performance over Pt/PG-0.12 for four cycles.

**Figure 5 molecules-29-01761-f005:**
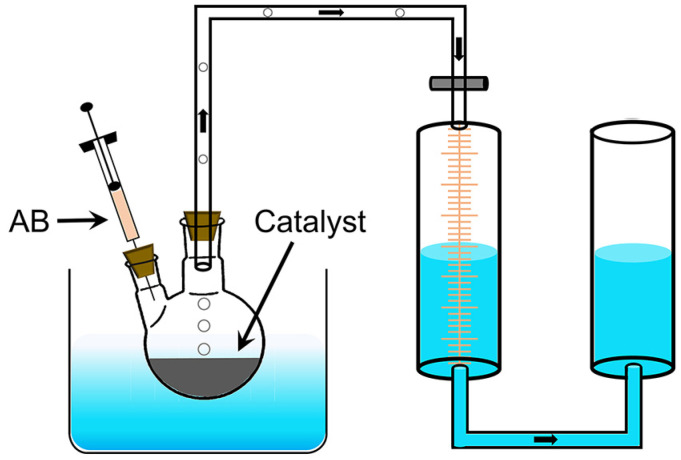
Schematic of catalysis device for the hydrogen release from AB hydrolysis.

## Data Availability

The data presented in this study are available on request from the corresponding author.

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
