# Peer review of "Exploring Enhanced Hydrolytic Dehydrogenation of Ammonia Borane with Porous Graphene-Supported Platinum Catalysts"

_molecules, 2024, doi:10.3390/molecules29081761_

Round 1

Reviewer 1 Report

Comments and Suggestions for Authors

The paper describes the procedure for the preparation of porous graphene supporting Pt nanoparticles which are subsequently used as catalysts for Hydrolytic Dehydrogenation of Ammonia Borane. The introduction reflects the interest of this paper and the objectives well described. The paper could be of interest for selected audience. However, I suggest some changes prior to its publication:

-The paper should describe in detail the catalytic device, including a new scheme or picture preferment, with special emphasis in the hydrogen measurement. Is it confirmed by any technique?

-In line 168, authors suggest the functionalization of the graphene at the basal planes. Could authors suggest what happen with graphene edges?

-The similarities or differences between the different Pt np in the different materials should be shown in detail. Also, is there any measurement of the Pt in each sample?.

-The discussion of the catalytic efficiency with the Pt should appear in the text.

Finally, in general, all the sub and superscript are not market in any section of the pdf version revised.

Comments on the Quality of English Language

Some expresions could be improved

Reviewer 2 Report

Comments and Suggestions for Authors

The work describes a catalytic system in which the porous material where the catalyst is trapped is based on reduced graphene oxide. The strategy adopted by the authors is to use sacrificial zinc oxide to insert nanopores into the material. The work is well written and readable, detailed in its characterizations and in my opinion does not require too many changes before it can be published in your journal. My opinion concerns minor revisions.

Abstract: The use of acronyms not explicitly indicated in the abstract is not recommended. please specify the term TOF (turnover frequency).

Experimental: Regarding the reproducibility of the experiments, fundamental data on sample preparation for different analytical techniques are lacking. Details on sample preparation for XPS and TEM/EDS, such as substrate, solution or solid deposition, etc. are missing. Details on the calibration of the ICP-MS instrument are completely missing. Authors are asked to enter the necessary information.

Results and Discussion:

The authors talk about “porous graphene”, but they should specify whether the material they obtain is a 2D laminar porous graphene or 3D conjugated interconnected porous structure.

Estimation of pore size by nitrogen adsorption can be biased by the presence of closed pores. If the authors were to also estimate the pore size via SEM, the article would undergo a truly significant improvement. The authors are asked to add some images to the SEM.

The effect of heat treatment at 300°C is unclear, given the subsequent treatment at 900°C. The authors should better justify their choice.

The conversion between the amount of Zn measured by XPS (0.83 at%) and its equivalence at 4 wt% is unclear (page 5 lines 199-201). Please write some more details.

The support file is useless, Authors can integrate the contents with the main text.

Round 2

Reviewer 1 Report

Comments and Suggestions for Authors

I am satisfied with the answers